# The Current Practice of Gradual Return to Work in Germany: A Qualitative Study Protocol

**DOI:** 10.3390/ijerph19063740

**Published:** 2022-03-21

**Authors:** Inga L. Schulz, Ralf Stegmann, Uta Wegewitz, Matthias Bethge

**Affiliations:** 1Department 3.5 Evidence Based Occupational Health and Workplace Health Management, Federal Institute for Occupational Safety and Health (BAuA), 10317 Berlin, Germany; stegmann.ralf@baua.bund.de (R.S.); wegewitz.uta@baua.bund.de (U.W.); 2Institute for Social Medicine and Epidemiology, University of Luebeck, 23562 Luebeck, Germany; matthias.bethge@uksh.de

**Keywords:** gradual return to work, rehabilitation, qualitative research, study protocol, occupational health and safety, workplace health management, workplace adjustments, documentary method of interpretation

## Abstract

(1) Background: The increase in working age and long-term illnesses is a challenge for society to maintain the health of employees and to support their work participation. In many countries, such as Germany, a gradual return to work (GRTW) is used frequently to support returning employees, but little is known about its facilitators and obstacles. This protocol aims to provide an overview of the national state of the art in workplace adjustments during return to work and to outline the design of a qualitative study examining current GRTW practices in Germany. (2) Methods: Our qualitative study will examine the current implementation of GRTW by means of 32 narrative interviews and 10 group discussions from different perspectives. Therefore, we will ask returning employees as well as organisational stakeholders and external experts about their experiences with GRTW and attitudes towards the measure. The verbal data obtained will be analysed using the documentary method of interpretation. (3) Discussion: This study will reveal facilitating and hindering factors for the implementation of GRTW in Germany. The findings of the study may contribute to an improved implementation of GRTW not only in Germany.

## 1. Introduction

### 1.1. Workplace Health Management and Return to Work

Work and its influence on health are subject to constantly changing influences, not least due to rationalisation or the extension of working life [1,2,3]. The increasing prevalence of chronic diseases and health-related disability pensions, the burden of long-term illnesses, as well as an ageing workforce and shortage of qualified staff are a growing public health concern worldwide, affecting not only individuals and companies, but also the economy and the healthcare system [1,2,3,4,5,6,7,8,9,10,11]. Thus, it is important to both maintain and restore the health, employability, and workability of the working population. Paragraphs 3 and 4 of the German Occupational Safety and Health Act as well as paragraph 618 of the German Civil Code (2013) secure the employer’s duty of care towards the employees with the aim of promoting their health at work and establishing preventive and humane working conditions. One way to achieve this is systematic and comprehensive workplace health management. Workplace health management is a strategy to protect, promote, and support employees’ health and wellbeing at work. Considering organisational concerns, resources and risk factors, an integrating approach of workplace health management interacts with occupational health and safety, operational integration management, workplace health promotion and personnel development [12,13]. On a meta-level, it describes a paradigm shift towards a salutogenic and holistic perspective, focusing on the employees’ potential and resources but also understanding companies as living organisms [14]. On a practical level, comprehensive workplace health management can, e.g., lead to the development of a value-oriented leadership. In Germany, this understanding also applies to the operational integration management (ger.: betriebliches Eingliederungsmanagement) which can contain instruments that help employees to return to work (RTW) sustainably [15,16], such as gradual return to work (GRTW) (ger.: stufenweise Wiedereingliederung).

### 1.2. Gradual Return to Work in Germany

#### 1.2.1. Legal Aspects

Law in Germany does not require the implementation of comprehensive workplace health management, whereas employers are obliged by law since 2004 to offer their employees operational integration management after 6 weeks of sickness absence (§ 167 Section 2 German Social Code IX, see [17] for more details). GRTW can be essential for the operational integration management process in Germany. However, it can also be initiated and carried out independently.

The operational integration management [18] and GRTW are anchored in Volume nine of the German Social Code. Both measures aim to promote the full, effective and equal participation in society of people with disabilities and those threatened by a disability as well as to avoid or counteract disadvantages (§ 1 German Social Code IX). Considering this legal and thematic context, GRTW is an independent transitional instrument [19] of tertiary prevention to specifically facilitate RTW after long-term sickness absence [17]. The key components are partially resuming the previous occupational activities at the original workplace and gradually adapting the work requirements (§ 44 German Social Code IX, § 73 German Social Code V, [20]).

GRTW in Germany is a voluntary measure for returning employees. It requires a prior assessment by a physician and the joint consent of the employee, physician, relevant social insurance institution, and employer. If employees are recovering in a GRTW measure, they remain on sickness absence. The measure is not legally binding for the employer unless the returning employee is severely disabled.

Any party involved can recommend or initiate GRTW. It usually lasts between 6 weeks and 6 months. During this period, the responsible social insurance agency is obliged to provide sickness absence benefits for the returning employee [21].

GRTW is based on a step-by-step plan or reintegration plan drawn up by the referring physician, ideally with the participation of all parties involved. In this plan, the duration of GRTW and slow increases of tasks in terms of time and content are presented in the form of stages [22].

#### 1.2.2. Purpose

GRTW as an integrating core measure [23] of RTW considers medical, therapeutic, job-related as well as individual factors and offers the opportunity to reconcile these while RTW. Serving therapeutic purposes, GRTW intends to positively influence the recovery and rehabilitation process. Therefore, an adjusted workload is recommended based on the individual conditions of the employees’ return and should be flexible, both in terms of time and content [24,25]. GRTW offers employees who are still unable to work, the opportunity to try out their work and stress capacity in the sense of training at an early stage [23]. Based on this, they can not only learn to assess their resources in a professional setting with the help of medical and therapeutic guidance but also increase their endurance and identify challenges that need to be addressed as well as reduce fears of relapse or excessive demands [23,24,26].

The outlined aspects underpin the therapeutic orientation of GRTW. Nevertheless, in the course of a long-term illness, it functions as a measure for processing and coping with the disease [27] at the interface between the individual, medical-therapeutic, rehabilitative, and operational levels [28]. Starting with identifying the employees’ needs through preparation, implementation, and after-care, various stakeholders and experts are involved in the transitional process of GRTW. This includes the treating physicians in clinics or private practices, who act on an external level. They determine the employees’ need for a GRTW and draw up, if necessary, in consultation with the treating therapist and social worker, an individual step-by-step or reintegration plan. During their return, employees who participate in GRTW may have contact with many different stakeholders. Each of these transitions in contact and flows of information represents an interface [29], thus posing a challenge for (gradual) RTW. Against the background of a comprehensive understanding of workplace health management, it is also central for a sustainable RTW that GRTW as a measure ideally is supported rather than hindered by other instruments, and possible interfaces (such as with operational integration management), as well as any synergetic effects, are addressed [30].

#### 1.2.3. Outreach

GRTW is the measure most frequently offered by employers in Germany within the framework of operational integration management [31]. German literature suggests that the total number of GRTW being recommended and carried out has steadily increased in recent years [20,32]. Although the instrument anchored in social law has been in practice since the 1970s, current research on gradual reintegration in Germany shows gaps.

A few published studies have shown that both the recommendation and utilisation rates of GRTW for treated patients vary considerably between rehabilitation clinics [32,33]. The recommendation rates made by the physicians differed considerably between 1.5–20% [32,34], depending on the study. Presumably, not only the physicians’ assessment of appropriate GRTW criteria is subjective and therefore heterogeneous [28,32] but clear indication criteria regarding the recommendation of GRTW is also lacking [33]. When looking at patients’ characteristics, sickness absence duration, as well as the productivity assessed by the doctor, have the most significant influence on the recommendation [32]. The supervisor’s attitude also plays an important role in the utilisation of GRTW, as the probability of the employee participating in GRTW increases if the supervisor shows interest and support in the continued employment of the returning employee [34]. An analysis of GRTW rates on behalf of the German statutory health insurances (GHI) by Schneider et al. [35] showed that only one-quarter of the employees who received a recommendation for GRTW participated in the measure. In small companies, the participation rate is even lower.

Younger employees are more likely to receive and accept an offer for GRTW, who, on average, have shorter periods of sickness absence and are already more likely to return to work [32,34,36]. GRTW participants in the study of Bürger and Streibelt [33] had lower rates in the intention of receiving a disability pension and, on average, showed a lower socio-medical risk than non-participants. This can be considered a positive selection of participants. Employees with long sickness absence periods (>3 months) and who have a correspondingly high risk of an early retirement benefit the most from GRTW on behalf of the German Pension Insurance (GPI) but do not receive the measure as frequently as they were supposed to, according to the authors’ empirical criterion [33,34]. However, there is an oversupply; in 20% of cases, the empirical criterion would not have required a decision in favour of GRTW [33].

GRTW after medical rehabilitation starts, on average, within six days after the end of treatment. Employees in outpatient clinics start the measure faster compared to individuals in inpatient rehabilitation [34]. The start of GRTW also depends on the type of disease the employees suffered from. While employees after orthopaedic rehabilitation treatment begin their GRTW the fastest, those in cardiac treatment start the latest [34].

Bethge [37] showed that about 16% of rehabilitants who attended an orthopaedic, cardiac, oncological, or psychosomatic rehabilitation participated in GRTW. Employees with GRTW were younger, more frequently female, and more severely restricted in working life [37]. A study conducted within a large German chemical company showed that 50% of all performed GRTW programmes were mainly due to mental and behavioural disorders, as well as musculoskeletal disorders [20].

#### 1.2.4. Implementation

GRTW is a flexible tool for returning to work regarding its legal requirements. The measure creates opportunities to tailor the RTW process individually to the needs of those involved. However, the current evidence indicates that its flexibility and individual orientation is used to a very limited extent, leading to the presumption that its potentials are neglected. Bürger et al. [34] found several indicators for an insufficient adaptation of the measure within individual cases. Other forms besides the increase of working hours are rare. Only 2% of the GRTW cases used a daily increase in workload [34]. GRTW plans were hardly ever used to adjust the increase in time and duties individually [34]. In one-third of all examined GRTW cases, returned employees judged the duration of the measure to be too short and the increase in duties and stress too fast [34,36]. More than half of the returning employees were dissatisfied with the support and care provided during GRTW, regardless of the disease. They wished for more assistance with health problems that may arise, professional questions, and organisational issues [36]. In addition to the lack of handling individual demands regarding work time, the authors identified a later start after the rehabilitation treatment and only one level of increasing tasks and duties as risk factors for an unsuccessful RTW [36]. In contrast, an orientation towards the individual needs of the employees and timely planning of GRTW seem particularly stabilising [23]. If these factors are considered, GRTW can minimise the employees’ fears regarding their RTW [23].

#### 1.2.5. Effects

Reintegration rates after GRTW are not recorded consistently in Germany. They vary between 75 and 90%, depending on the illness, provider, or company [20,28,36,38,39]. It is not always clear how sustainable RTW is.

Streibelt et al. [38] pointed out that 88% of employees with chronic mental disorders that were supplied with GRTW sponsored by the GPI attained a full RTW, but only 73% of those attained this status without GRTW. With GRTW, the risk of becoming unemployed and receiving a disability pension fell by 60% [38]. Sickness absence within 15 months after completing GRTW decreased by six weeks compared to employees returning without GRTW. In addition, GRTW showed positive effects on the employees’ subjective health ratings [38]. Gradual work resumption seems to be more beneficial for employees with an uncertain or negative subjective RTW prognosis.

Bethge [37] showed a decrease of almost 40% in entering disability benefits for employees who returned to work gradually after various chronic diseases [37]. Bürger and Streibelt [40] concluded from their research on GRTW provided by the GPI that after completion of psychosomatic rehabilitation, employees derive the greatest benefit from participating in GRTW, followed by GRTW after musculoskeletal rehabilitation. In oncology, as well as in cardiology, no additional benefit was identified [40].

The only study examining GRTW using data from the largest GHI presents the advantages of participating in GRTW on behalf of the GHI for individuals with sickness absence above 120 days [35].

Globally, different concepts exist to handle RTW, but many follow the same idea as GRTW for sick-listed employees in Germany. These concepts can, for example, include part-time or partial sick leave, light duties, work accommodations or work adaptions, graded activities, phased RTW, graded work exposure, or modified work. International literature suggests that the gradual resumption of work after a long period of sickness absence is effective in getting employees back to work, keeping them at work, assuring their health, and reducing costs arising from prolonged sickness and loss of productivity [15,16,41,42,43,44,45,46,47,48,49,50,51]. Next to other variables that affect the process of RTW, the current research and literature lead to the conclusion that a sustainable RTW is more likely when GRTW is facilitated [47,50,52,53,54,55,56].

## 2. Methods and Design

### 2.1. Aim

From a research point of view, GRTW in Germany is currently sparsely examined (see Discussion for more). While studies have shown that the full potential of GRTW is currently not being realised, it is not clear what might cause these deficits. In addition, little is known about the practical implementation, barriers, and facilitators of GRTW, even though this knowledge is important for successful GRTW as well as to tap into the full potential of GRTW in Germany.

Our study aims to gain information and knowledge from different perspectives about the current process and practices of GRTW in Germany. Our research systematically focuses on the implementation of GRTW in Germany, considering the given possibilities and taking into account the limits of the measure. We will examine current GRTW practices and experiences of affected employees and the stakeholders involved, as well as the technical, operational, and experiential knowledge on which these practices are based. The case consultations of the individual interviews and the additional group discussions will offer the opportunity to explore multiple perspectives on the process (see Figure 1). The interviews will uncover the GRTW practices, expectations, and needs through both the employees’ experiences and the company-associated trusted persons’ perspectives. Additionally, existing barriers to the implementation and determinants for the success of GRTW as a policy instrument will be identified in the group discussions. On this basis, the results will be available for explanatory approaches, as well as action recommendations and practical tools (e.g., requirement criteria catalogues, recommendation routines, and guidelines for the creation of demand-oriented step-by-step plans). To the best of our knowledge, this is the first study of its kind.

### 2.2. Research Questions

We want to explore how affected employees, organisational stakeholders as well as external experts experience and describe GRTW. In more detail, we will clarify the following three questions:What experience and action-guiding knowledge do the affected employees, organisational stakeholders as well as external experts contribute to the planning and implementation of GRTW? How does this influence RTW in general?What do affected employees, organisational stakeholders as well as external experts experience as beneficial or as hindering within the GRTW process, and why?How do affected employees, organisational stakeholders as well as external experts describe and experience the decision-making within the GRTW process (e.g., GRTW design; step-by-step plan; underlying illness)?

### 2.3. Study Design

The study follows an explorative qualitative research approach. It aims to provide detailed insights into the GRTW practices in Germany from multiple perspectives. Qualitative research has proven its worth for investigating action practices and implicit knowledge [57,58]. The verbal data gathered through interviews and group discussions allow the reconstruction of the realities of the participants’ experiences with GRTW [59].

The qualitative data will be collected in two study arms (displayed in Figure 1). In the first study arm, we will interview 12 returning employees individually, shortly before the start of their GRTW and three months later. Parallel to the three-month follow-up interviews, we will additionally interview eight organisational trusted stakeholders from the company who have accompanied the GRTW process of the questioned employees.

In the second arm of the study, we will interview returned employees, organisational stakeholders, and external experts through 10 group discussions.

Thus, the present study will exploit the advantages of two different qualitative survey methods: narrative interviews and group discussions, both using guided questions as support.

The potential of the narrative interviews can be identified in the underlying narrative theory (ger.: Erzähltheorie) [60]. Given the researcher’s greatest possible openness and a narrative-generating impulse, the respondents are free to structure their statements within the framework of the research topic ‘GRTW’. The respondents consistently set the focus, as well as the beginning and end of the narrative [61]. We will not only ask for opinions and everyday theories of the respondents but also try to elicit narratives about personal experiences with GRTW, such as those experiences that are sound in the action practice [58]. Since the study aims to record the current GRTW practices from multiple perspectives, we perceive this method to be appropriate.

Group discussions, however, benefit from the emerging dynamics and self-running discourse, in which important collective experiences are addressed by the group [62].

By referring to different (especially contrary) views of the other participants, individual opinions and experiences on GRTW are expressed not only more spontaneously but also more clearly [63]. Thus, group discussions enable access both to a collective stratification of experience and to conjunctive contexts of origin for collective orientations in the field of RTW [64] in the sense of conjunctive experiential spaces (ger.: konjunktive Erfahrungsräume) [63].

Using the documentary method of interpretation, both group discussions and interviews provide access to not only intentional practices, reflected views, and explicitly available expertise but also to preconscious action routines and implicit knowledge of actions and experiences within the field of RTW [58,63,65]. Methodologically, group discussions may complement the narrative interviews as part of an across methods triangulation [65] to understand the phenomenon of GRTW in full depth [66]. We will look for possibly different but complementary information to obtain a bigger picture of the research subject. The combination of these two methods allows us to investigate GRTW comprehensively from multiple perspectives based on individual examples and collective orientations.

### 2.4. Sample Profile

The study population will consist of different target groups from the RTW context. All target groups must have experiences with GRTW to some extent that is referring to the research questions. We will aim to include as many individuals experienced with GRTW as possible in the present study population. In terms of field development and ‘nosing around’ [67] during the qualitative research process, the previously defined study population may change. In this context, it should be mentioned that during field development, first contacts with experts will be established, and information will be exchanged. In this way, we will gain new insights that can reflexively drive the future research process. Thus, it is possible that the participation of specific individuals or groups of persons will suddenly appear particularly relevant. Therefore, to a certain extent, the study population’s definition will remain flexible.

All participants must be able to take part in a conversation and show adequate German language skills. At the current state of knowledge, the individuals and groups of persons will be included in the study population as follows:

#### 2.4.1. Interviews

(a)12 returning employees who are planning to return to work using GRTW after a period of sickness absence and between 18 and 60 years old.(b)Eight trusted persons from the same company of the interviewed employees who were proposed by the returning employee and who have accompanied GRTW of the employee referred to above.

#### 2.4.2. Group Discussions

Four organisations or companies in Germany that differ in terms of company size, sector, and location will be included in the study. The study population will arise from the same company so four pairs each will be assigned to an organisational group discussion.

(a)Four group discussions with three to five returned employees between 18 and 60 years old, who have returned to work via GRTW after a longer period of sickness absence at least once in the company they are currently employed at within the last three years of employment.(b)Four group discussions with three to five organisational experts and stakeholders who have long or intensive professional experience in the field of GRTW or RTW.(c)One group discussion with four to six external experts and stakeholders from GPI, GHI, integration services, and other services that support RTW and who have professional experience in the field of GRTW and/or RTW in their current position.(d)One group discussion with four to six external experts and stakeholders from medical and/or therapeutic healthcare, social work, occupational medicine, rehabilitation, and acute clinics.

### 2.5. Sample Selection and Recruitment Process

The contrasting sampling procedure in this study is based on theoretical sampling [68]. Thus, recruiting will be an ongoing, accompanying process [67]. We will successively select the study participants based on the criteria mentioned above to ensure heterogeneity [67]. The procedure aims to represent the heterogeneity of the research field at least rudimentarily through conscious case selection [69]. The case selection will be based on continuous comparison of the cases, which highlights the minimum and maximum contrasts following the criterion of proven theoretical relevance [68]. Sampling will be completed when the resulting theoretical concepts are saturated.

Regarding recruitment, we have developed a study flyer that can be easily accessed and will be distributed by multipliers both for the interviews and group discussions.

Financial reimbursement will be offered to all participants to compensate for the time and effort spent.

#### 2.5.1. Interviews

At measurement point one, we will interview 12 returning employees to ensure that at least eight study participants who finish their GRTW can be interviewed twice. Returning employees who discontinue their GRTW are not excluded from the study. These cases will be used as contrasts and can also provide meaningful information about the reasons for failure and the need for optimisation of the measure.

With the help of gatekeepers, i.e., orthopaedic, psychosomatic, oncological, and cardiac rehabilitation and acute clinics, general practitioners’ practices, and companies, potential study participants will be reached. The participating employees, in turn, will act as gatekeepers and will enable contact with the organisational stakeholder who accompanied their return. This contact will occur after the first interview and before the second survey time.

The cooperating clinics, medical practices, and companies will be located in different regions of Germany to ensure that the study participants will be from different parts of Germany. The sample will be compiled as diversely as possible concerning the following criteria: age, gender, diagnosis, place of residence, and GRTW after rehabilitation versus without previous rehabilitation.

#### 2.5.2. Group Discussions

For the recruitment purpose, we will use existing contacts from previous studies at the Federal Institute for Occupational Safety and Health. Furthermore, we will contact potential study participants directly at events by e-mail. Existing organisations, associations, and federations, as well as their distributors, will function as multipliers.

Doctors, therapists, and social workers of the cooperating clinics from the first study arm and other experts will participate in the group discussion with medical and therapeutic experts.

We will select the companies from various industries within different German regions. The aim is to achieve the greatest possible heterogeneity in terms of location, size of the company, and sector. Other participants in the group discussions will be recruited via an organisational contact person acting as a gatekeeper, such as disability managers or RTW coaches, who know the company structure and the returning employees they may approach.

### 2.6. Data Collection

We will develop the guiding questions for the interviews and group discussions based on the current literature and experience from previous qualitative studies at the Federal Institute for Occupational Safety and Health. The largest part of the guiding questions will be formulated to be as open as possible to trigger detailed narratives and descriptions. The second part will contain more explicit questions concerning personal opinions and assessments of both GRTW and RTW.

We will include questions referring to experiences with and attitudes towards GRTW, as well as its role in the process of RTW. The guides will be adjusted and tailored to address the different target groups of the study appropriately. A total of eight separate interview guides will be developed for the interviews (four) and group discussions (four). We will jointly prepare the guides and reflect on them in our team of researchers at the Federal Institute for Occupational Safety and Health. Using the feedback of two pilot interviews with returned employees, the questions will be revised in a participatory way. The interview and group discussion guides are available from the corresponding author upon request.

In addition to the narrative interviews and group discussions, the participating employees, organisational experts, and stakeholders will receive a short questionnaire in both the individual interviews and group discussions. These questionnaires will be designed to describe the sample in more detail and gather information about the target group, which will contribute to the final case or discourse description. The short questionnaires will contain questions regarding age, gender and GRTW, but also regarding the self-reported workability [70], the current health status [71] and RTW self-efficacy [72].

The first author will conduct the interviews and moderate the group discussions. At the beginning of data collection, an experienced researcher (second author) of the institute will both support the interviewer and give feedback. The participants will be informed about the aim of the study, the procedure, data privacy, and the professional background of the research team and interviewer prior to the data collection in written and verbal form (see Ethics approval and consent to participate). Compared to the first interviews (t1) which will last 60 min and the second interview (t2) which will last 45 min, the group discussions will take 120 min. The first interview with the employees (t1), as well as the group discussions, will take place in person, e.g., at the rehabilitation clinic, at home, at the company, or at the Federal Institute of Occupational Safety and Health in Berlin. At the second measurement point (t2), the employees, as well as the associated trusted person, will be interviewed by telephone. Both the interviews and group discussions will be audio-recorded entirely for later transcription. An external provider will transcribe the data.

To keep the researchers’ bias to a minimum, the researchers will maintain written records of their theories, assumptions, and impressions in a research diary. These notes will be recorded throughout data collection and will be reflected on.

As soon as the data has been evaluated, we will present the results to some of the study participants in a workshop, including at least one representative of each category of individuals met. This procedure has already proven itself as sufficient in our previous studies to stimulate discussion about the transfer of the results and recommendations for action.

### 2.7. Analysis

Analysis and data collection were run in parallel by means of a circular research process and regarding theoretical sampling. We analysed the verbal data obtained by using the documentary method of interpretation according to Ralf Bohnsack [63]. It is a method of reconstructive social research that can be used for both interviews and group discussions [63]. The interpretation took place in the sequential reconstruction of narrative, interaction, and discourse processes [73].

According to Bohnsack, the method focuses on both levels of discourse, but goes beyond the literal or immanent meaning (by asking what) and asks for the documentary meaning, the pre-reflexive, implicit, or tacit knowledge (by asking how) [73]. The how aims to reveal in which framework the topic is dealt with, which is also called the framework of orientation [63]. Using the documentary method, the researcher “[…] is able to find an access to the structure of action and orientation, which exceeds the perspective of those under research.” [74] (p. 101). As stated by Bourdieu [75], the term structure of practice refers to the habitus or the modus operandi of everyday activities. The various evaluation steps (Figure 2) help with understanding (ger.: ‘Verstehen’), according to Mannheim [76], the action-guiding knowledge gained from previous experiences with GRTW that is evident in everyday practice and study participant’s activities. Subject to interpretation within the documentary method are the frameworks of orientation and patterns of meaning by comparing other cases or groups [74]. The task of the method is to explicate this implicit knowledge [74]. Regarding group discussions, Bohnsack [74] highlights: “[…] it is above all the (formal) organisation of discourse which has to be reconstructed. This means we had to characterise the way of how participants refer to each other formally in their utterances” (p. 111). This statement underlines the difference between interviews and group discussions clearly; in interviews, the participants set their own relevance and foci, while in group discussions, the reciprocal reference between the participants, the discourse, and the joint consensus were analysed equally.

The documentary method is also a comparative analysis procedure that, using contrastive case comparison, leads to cross-case findings, i.e., type and theory formation. We visualised the individual analysis steps in Figure 2.

The researchers will structure the material by means of sequences and themes of GRTW in a thematic course [77]. In this step, we will select passages that are relevant to the research questions and include them in the following interpretation. Thematic courses can be seen as the first part of the formulating interpretation [77]. In this more detailed evaluation, the material will be sequenced according to the main topics and subtopics [77]. The following step of interpretation, reflective interpretation, will be strictly separated to represent the differences between immanent and documentary meaning [58,77]. First, the narrative text sequences will be analysed formally regarding their text types or genres and homologous patterns (e.g., proposition, elaboration, conclusion) [58,73,77,78]. Second, the semantic level of interpretation will be targeted by comparing the framework of orientation in which the topics or problems are elaborated. Thereby, atheoretical, implicit knowledge will emerge.

Furthermore, the subsequent comparative analysis is of great importance for this reconstruction [63,79]. The case with its particularities, as well as the overall shape of the case, will be relevant reference points for the case description of the interviews and discourse description of the group discussions [79]. It will contain a presentation and condensation of the interpretations and results [79]. We will incorporate all the data collected into the descriptions by means of interviews or group discussions and short questionnaires. As an integral part of the documentary method, cross-case comparisons (person A versus person B), as well as case-internal comparisons (t1 versus t2 of person A), frame the whole process of analysis. During the progress of interpretation and comparison, the orientation patterns and frameworks of GRTW will become more and more explicit. Moreover, the process of analysis by means of the documentary method will conclude with the step of type formation [78,80]. In this research project, we will strive for a sense-genetic type formation and look for topics or problems all cases or groups have in common [80]. Using the sense-genetic type formation, we will show similarities and differences of the orientation frameworks in which the study participants deal with topics and problems that focus on GRTW [80].

## 3. Discussion

As shown above, the current research literature in Germany is dominated by studies that have examined GRTW under the responsibility of GPI. Thus, it can only give insights into a small amount of GRTW programmes performed after the completion of rehabilitation treatment in Germany. The informative value is therefore limited since the results from GRTW carried out by the GHI cannot be transferred without restriction. Apart from this, the included studies from Germany used a quantitative approach throughout to investigate GRTW. Qualitative research focusing on expectations, decision making, planning, and implementation of GRTW in Germany is missing.

The studies conducted to date, nevertheless, provide initial indications of deficits in the access, planning and process quality of GRTW and their possible causes. Differences resulting from company size or workplace-related characteristics, such as shift work, however, are hardly taken into account. The lack of such studies leads to a gap in valuable scientifically proven insights and findings that could in return guide actions to reduce deficits in access and process quality of GRTW. Of course, it must be taken into account that GRTW interventions at the workplace vary widely from country to country and are therefore only partly comparable to the German version of GRTW.

The strengths of the study lie in the multiple perspectives included and the qualitative approach regarding triangulation to shed light on current practices and experiences with GRTW, as well as on the barriers and facilitators of GRTW, from different points of view. The individual interviews at two measurement points will accompany the process of reintegration and give insights into the perspectives of both the returning employee, comparing expectations and the actual implementation of GRTW, and the selected trusted person in the company. The group discussions will provide insights into the operational and systemic structures, as well as dynamics of GRTW and collective orientations, on a retrospective basis.

Due to the ongoing COVID-19 pandemic, we expect weaknesses in the recruitment of study participants and implementation of the surveys. It will not be possible to conduct all interviews and group discussions personally. Instead, video conferences and telephone interviews will be used. Therefore, we will reflect on differences in the type and manner of the surveys and, if necessary, in the analysis. In addition, the effects of the pandemic could result in fewer GRTW programmes, making it more difficult to find enough study participants. Furthermore, it must also be considered that GRTW that takes place during the COVID-19 pandemic may be carried out under very different circumstances than usual, such as a home office or limited contact with team members. Group discussions conducted retrospectively could possibly control for this effect.

## 4. Conclusions

The study will not only reveal barriers and potentials of the recommendation, planning, utilisation, and implementation of GRTW in Germany. It will also provide explanations for current problems and hindering determinants. The results can also provide information on how GRTW is currently embedded in the process of operational integration management and what potentials arise from it. Besides that, the study results can also help to better integrate GRTW in a holistic approach to workplace health management in the future. In doing so, however, the limits of GRTW should also be considered in order to use and implement the measure in an effective and satisfactory manner for all those involved.

## Figures and Tables

**Figure 1 ijerph-19-03740-f001:**
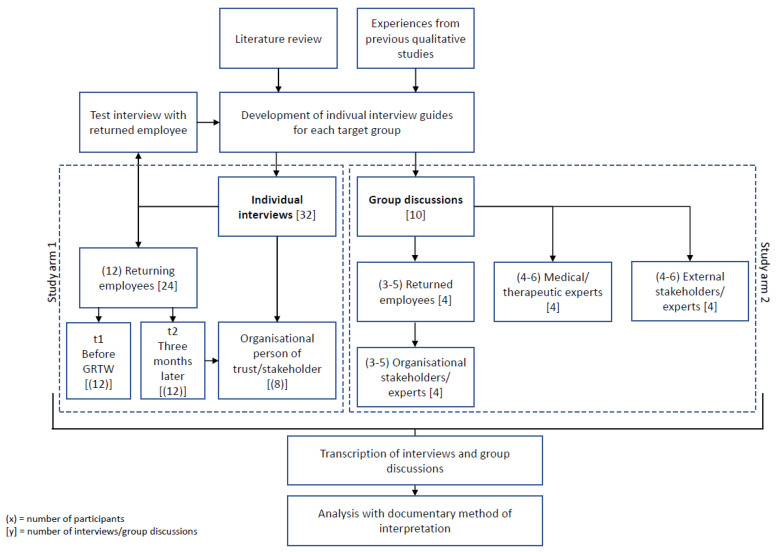
Planned data collection and its embedding.

**Figure 2 ijerph-19-03740-f002:**
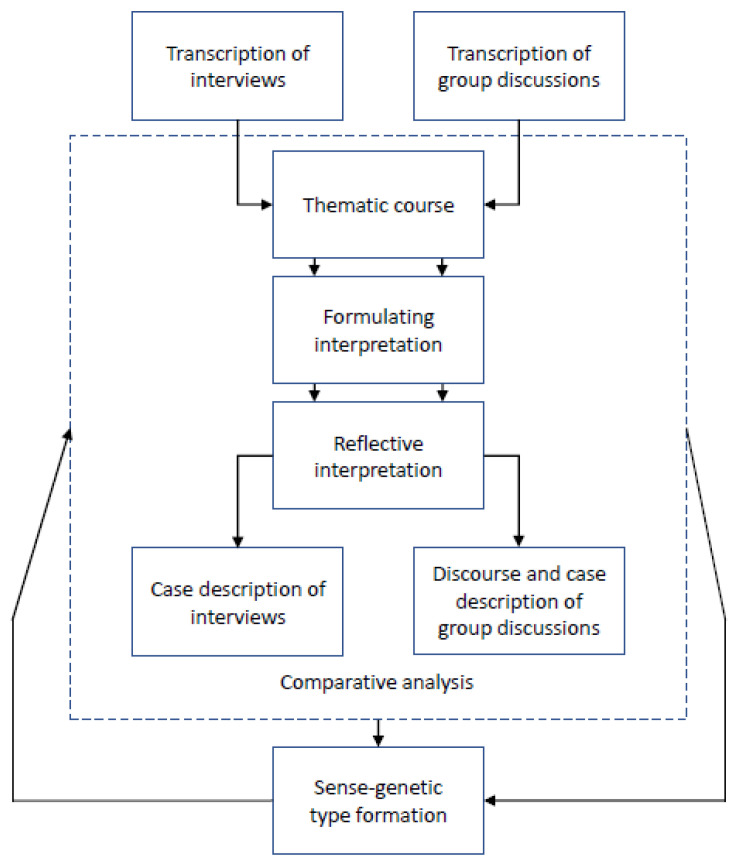
Exemplified steps of analysis within the documentary method of interpretation.

## Data Availability

Interview and group discussion guides are available from the corresponding author upon request. Other data sharing is not applicable to this article as no datasets were generated or analysed yet.

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
