# Peer review of "The Current Practice of Gradual Return to Work in Germany: A Qualitative Study Protocol"

_ijerph, 2022, doi:10.3390/ijerph19063740_

Round 1

Reviewer 1 Report

Thank you for inviting me to review this manuscript.

I have a better understanding now of return to work in Germany. The manuscript is well written with my comments mainly about grammar. See below for specific comments but also it would be worth having the manuscript reviewed by a native English speaker.

Abstract;

Line 19: 10 group discussions 19 from different stakeholder perspectives

Line 20 – 22:

Change this sentence to: “Participants will include returning employees as well as 20 organisational stakeholders and external experts about their experiences with GRTW and attitudes toward the program”

Q: what are supra-organisational experts? please find another term that is more easily understood internationally. Perhaps, ‘external’

Q: toward the measure of what?

Line 24:   hindering factors for the implementation

Introduction

The Introduction is 4.5 pages long. While it is an interesting read, I think it could be written more concisely. GRTW is not a new concept and has been extensively researched internationally.

Line 43- 44: Rewrite this sentence to :Paragraphs 3 and 4 of the German Occupational Safety and Health Act as well as Paragraph 618 of the German Civil Code (2013) secure the employer's duty of care towards the employees with the aim of promoting their health at work and to establish preventive and humane working conditions.”

Line 47 – remove ‘a’

line 39: change ‘integrating’ to ‘integrated’

line 52: what does ‘it’ refer to?

line 55 – 57: On a practical level, a comprehensive workplace health management can for example, lead to the development of a value-oriented leadership.”  It is unclear if ‘workplace health management’ an approach or framework or plan. Could the authors please clarify this phrase? Also, please remove ‘a’ before ‘workplace health management’. To me, it reads like a plan or approach that an organisation will take to support employee health and wellbeing.

line 74 – remove ‘a’

line 83: what does ‘both measures’ refer to?

line 91: it is confusing to call GRTW a ‘measure’. Perhaps the term ‘strategy’, ‘program’ or ‘intervention’ is more accurate. Also Line 95 and throughout the manuscript.

Line 101: what is GRTW a ‘so-called’ approach. The phrase ‘so-called’ suggests that the authors don’t agree with the step by step approach. Please clarify.

Line 160, 166: I don’t think the word ‘rehabilitants’ exists in English. Please replace.

Line 297: Remove the question mark

line 458: the word should be ‘therapy’ not ‘therapeutic’

Lines 484-5: Regarding this sentence, have the interview and group discussion guides been established? If so, they can be added as supplementary data to this manuscript.

There is no mention of participant reimbursement. Please comment.

Well done a comprehensively and clearly written paper.

Reviewer 2 Report

Congratulations to the protocol esablishment!

It sounds reliable and useful and helpful for future RTW processes.

The only concern is the number of the participants (12 + 6) and I am very much in doubts that only this study will bring a sufficint scientific approval. more or less it is a pilot? 

So my question was if it was ment to be a pilot study or how they
would increase the number of participants to prove the validity of
the research.

Reviewer 3 Report

The manuscript explores gradual return to work (GRTW) as an instrument in Germany and shows the study protocol how to qualitatively evaluate it in Germany. 

The main strengths are comprehensive literature regarding current situation of GRTW in Germany and in-depth methodology of the study protocol. 

General comments: 

The literature review in Introduction is really comprehensive. I find it maybe too comprehensive regarding legal aspects etc. but I think some can have use of this information. 

I find it difficult to find the clear justification to this study and study protocol. Aims are provided in 2.1 but it feels that the justification is provided more extensively in the beginning of the Discussion. I would clarify the justification for this project (earlier than in Discussion) and current limitations in this study field. 

What is the rationale for interviewing 12 individuals? Considering the rate of failure of an employee to return to work, is the amount appropriate? I didn't find any discussion about this. 

During covid pandemic, should the methodological issues be considered already in the study design instead of just discuss this issue in the limitations section?

Specific comments:

row 60: please open the abbreviation RTW.

row 173: please open the abbreviation GHI.

row 185: please open the abbreviation GPI.

rows 233-34: RTW and return to work is not used systematically 

2.2 Research questions: I would avoid using 'and' so much and replace some of those with the comma. 

row 361: between (or) across, please clarify

row 405: Is the three-year period sufficient? One could think that three years old practical issues would be pretty hard to recall?

2.4.2: c) and d) reports four to six experts/stakeholders, Fig 1 reports 1x8. Was there something I didn't understand?

2.5.1: Is the company size been considered in study population criteria? 

rows 484-5: I would consider rephrasing the sentence. 

rows 505-7: how do the authors ensure that employees would be from different parts of the Germany, considering face-to-face interview? So that the study population would not be too homogeneous (only near Berlin). 

Reviewer 4 Report

Dear authors,

I can only congratulate you for the enormous work you are doing. If some methodological aspects raise questions, they will be answered in the publication of the associated articles. We regret that this study does not use a mixed research methodology, putting the emphasis on the qualitative ... even if it means having a smaller quantitative sample.
Some parts in the methodology are difficult to read and understand. One has to go over it several times to understand the interest of each step. The figures are not so clear either.
If I had had to write this project, I would have proposed a final validation step of the results with a focus group including at least one representative of each category of individuals met.

Good luck with this huge job.
